# Efficacy of different courses of acupuncture for diarrhea irritable bowel syndrome: A protocol for systematic review and meta-analysis

Junjian Tian[1]☺, Ting Li[1]☺, Jun Zhao[1], Da Li[2], Jingwen Sun[1], Zhigang Li[1]*, Rongxing Shi[3]*

1 School of Acupuncture-Moxibustion and Tuina, Beijing University of Chinese Medicine, Beijing, China, 2 Beijing University of Chinese Medicine Affiliated Dongzhimen Hospital, Beijing, China, 3 China-Japan Friendship Hospital, Beijing, China

☺ These authors contributed equally to this work.
* lizhigang620@126.com (ZL); 2243111436@qq.com.cn (RS)

**Data Availability Statement:** No datasets were generated or analysed during the current study. All relevant data from this study will be made available upon study completion.

## Abstract

Irritable Bowel Syndrome (IBS) is the most common functional gastrointestinal disorder. As one of the most common subtypes of IBS, IBS-D can impair the patients' quality of life (QOL) and decreased work productivity. Acupuncture may be a potential treatment for patients with IBS-D. However, the treatment course of acupuncture was diverse. It is unclear what is the optimal acupuncture treatment courses for acupuncture. The efficacy and safety of different courses of acupuncture for IBS-D have not been systematically evaluated yet. The purpose of this study is to evaluate effectiveness of Acupuncture of different courses in the treatment of IBS-D and provide sufficient evidence for clinical recommendations for IBS-D. We will follow the Preferred reporting items for systematic reviews and meta-analysis protocols (PRISMA-P) guidelines to design the protocol of a systematic review and meta-analysis. This systematic review is registered in PROSPERO (CRD42023418846). We will include randomized controlled trials (RCTs) in which the efficacy of Acupuncture is compared with a placebo, sham acupuncture or Pinaverium bromide in the treatment of IBS-D with no language restrictions. The outcomes of interest will be efficiency rate and the Symptoms Severity Score. RCTs will be searched in the electronic database and Clinical Trials Registry Platform from inception to April 2023. Two independent reviewers will independently select studies, extract data from the included studies, and assess the risk of bias using the Cochrane tool. We will choose a random or fixed-effects model based on the heterogeneity index. We will use the relative risk and mean difference to estimate data with 95% CI. A stratified meta-analysis was conducted to evaluate the effect of different treatment courses of Acupuncture: 2weeks, 4weeks(or 1 months), 6 weeks, and 8 weeks. If there is significant clinical and methodological heterogeneity, we will look for the reason for heterogeneity and perform a subgroup analysis. According to the Grading of Recommendations Assessment, Development and Evaluation (GRADE), we will evaluate the evidence quality and provide the recommendation's strength.

**Funding:** This research was supported by The National Key Research and Development Program of China (Project No. 2019YFC1709004, Financial No. 2030071420057).

**Competing interests:** The authors have declared that no competing interests exist.

## Introduction

Irritable bowel syndrome (IBS) is a common functional disorder of gastrointestinal system. The total prevalence of IBS among adults is between 4.6% and 9.0% worldwide [1]. According to the main stool pattern characteristics, IBS can be classified as diarrhea-type IBS-D, constipation-type IBS-C, diarrhea-constipation-mixed IBS-M, or unclassified IBS-U [2]. IBS-D is the most prevalent subtype [3], accounting for approximately 31.5% of IBS [4].

Although the pathogenesis of irritable bowel syndrome (IBS) is complex and still unclear [5], evidence indicates that its pathophysiological changes is associated with a dysregulation of brain-gut interactions [6].

Disrupted gut motility, visceral hypersensitivity, changed mucosal and immunological function, and modifying brain connections and activity are all relevant causes [5].

Patients with IBS usually suffer from abdominal pain and discomfort associated with altered bowel movements [7], which may considerably impair the patients' quality of life (QOL), decreased work productivity [8], and increased healthcare costs [9].

The severity of IBS symptoms is an important factor affecting work and efficiency [10]. Among those who are out of work within a week, 2.9–4.3% are for irritable bowel syndrome [10]. A recent meta-analysis based on 11 European countries, involving more than 2,700 patients, showed that each patient associated with IBS costs about 3,000 euros per year [11]. In the United States, the previously estimated total direct cost of managing IBS was more than $1 billion a year, while the indirect cost associated to productivity loss was over $200 million [12].

Effective treatments for IBS are needed to relieve symptoms, improve quality of life, and to reduce healthcare utilization [13–15].

Initial treatment for IBS includes lifestyle modification, pharmacotherapy, and complementary and alternative therapie. With recurrent disease and long-term side effects, medication can only relieve symptoms and does not significantly improve patients' quality of life (QOL) [16]. In addition to drug therapy, complementary alternative therapies can also play an important role in the management of these patients. Approximately 50% of patients with IBS [17] sought alternative therapy after conventional treatment failed. In recent years, Acupuncture as a complementary and alternative therapie has been widely selected as one of the treatment strategies for IBS-D patients [18, 19].

According to the comprehensive review, acupuncture therapy may relieve IBS-D symptoms by regulating brain-gut peptides, modifying cerebral connection and activity, and reducing inflammation and intestinal hypersensitivity [20].

Acupuncture in the alleviation of IBS symptoms is superior to Pinaverium, and adverse events are lower than most antispasmodics [21]. Acupuncture treatment can improve the clinical effectiveness of IBS-D, with great safety [22]. The results of the systematic review and meta-analysis reveal that acupuncture could improve symptom severity, abdominal discomfort, and quality of life in people with IBS [23].

However, the acupuncture treatment courses in the included RCT varied. Unfortunately, the optimal acupuncture treatment courses is still unclear. Up till now, few systematic reviews have been conducted on the efficacy and safety of acupuncture to different treatment courses for IBS-D. Therefore, this study will summarize the existing evidence and conduct a meta-analysis to evaluate the efficacy and safety of different courses in the treatment of IBS-D, and to provide a more accurate and reliable resource for the clinic.

## Methods

We will follow the Preferred reporting items for systematic reviews and meta-analysis protocols (PRISMA-P) guidelines to design the protocol of a systematic review and meta-analysis

[24, 25]. The review protocol was preregistered at the International Prospective Register of Systematic Reviews (PROSPERO) under the registration number CRD42023418846.

## Eligibility criteria

**Inclusion criteria.** *Participants.* The patients diagnosed with IBS-D will be eligible based on Rome IV/III/II criteria [26–29]. Participant characteristics such as age, gender, country of origin, will not be restricted.

*Interventions.* The interventions will include acupuncture or EA. The electrical stimulation intensity, acupuncture depth, the number of acupuncture points and treatment courses will not be restricted.

However, studies in which acupuncture is combined with other conventional treatments should be excluded.

*Comparators.* The comparison groups will receive placebo, sham acupuncture or Pinaverium bromide treatment. And studies comparing merely various acupuncture prescriptions or acupuncture types will be excluded.

*Outcomes.* The primary outcome measures of this review will be the efficiency rate and the Symptoms Severity Score [30]. The clinical efficiency rate was defined as the degree of relief of IBS symptoms, which are assessed through the Symptoms Severity Score. This includes the use of the IBS Symptom Severity Scale (IBS-SSS) [31], or the Gastrointestinal Symptom Rating Scale for IBS (GSRS-IBS) [32].

Secondary outcome measures will include abdominal pain scores, abdominal frequency scores, satisfaction with bowel habits scores, quality of life [33], and the total incidence of adverse effects.

The ratio of reported adverse events to the total number is used to evaluate safety.

*Type of studies.* We designed this protocol to consider all RCTs that evaluate the efficacy of different courses of acupuncture for IBS-D. There are no language restrictions on the studies that meet these criteria.

**Exclusion criteria.** The exclusion criteria include literature reviews, animal experiments, single case reports, non-randomized controlled trials, observational studies, and unfinished protocol.

## Search strategy

RCTs will be searched from inception to April 2023 in the following databases: Embase, PubMed, Cochrane Library, Web of Science, and four Chinese databases (China National Knowledge Infrastructure, China Science and Technology Journal Database, WanFang database, and China Biology Medicine disc). Articles will also be searched from the International Clinical Trials Registry Platform (ICTRP), the National Institutes of Health (NIH) clinical registry Clinical Trials, and ClinicalTrials.gov. The search terms will include "Acupuncture," "Acupuncture Therapy," "Acupuncture Points," "irritable bowel syndrome," and "IBS." We will search the databases using a combination of the subject terms and free words. A description of PubMed's search strategy can be found in Table 1.

**Data management and study selection.** All electronically searched articles will be exported to EndNote X.9 software. If there are duplicates, we will include only one piece of literature. Two independent reviewers will screen potentially relevant studies based on the inclusion and exclusion criteria. In the first stage, nonrelevant trials will be excluded by reading the title and abstract. Then two independent reviewers will independently assess the selected studies. If the studies meet the requirements, they will be included in the systematic review. When two reviewers have discrepancies, we will invite the third reviewer to decide if the study is to

**Table 1. Search strategy of PubMed database.**

| | Search items |
|---|---|
| #1 | ("Irritable Bowel Syndrome"[Mesh]) |
| #2 | (((((((Irritable Bowel Syndromes[Title/Abstract]) OR (Syndrome, Irritable Bowel[Title/Abstract])) OR (Syndromes, Irritable Bowel[Title/Abstract])) OR (Colon, Irritable[Title/Abstract])) OR (Irritable Colon[Title/Abstract])) OR (Colitis, Mucous[Title/Abstract])) OR (Colitides, Mucous[Title/Abstract])) OR (Mucous Colitides[Title/Abstract])) OR (Mucous Colitis[Title/Abstract]) |
| #3 | #1OR#2 |
| #4 | ("Acupuncture"[Mesh]) |
| #5 | ((((((((((((((((Pharmacopuncture[Title/Abstract]) OR (Acupuncture Therapy[Title/Abstract])) OR (Acupuncture Treatment[Title/Abstract])) OR (Acupuncture Treatments[Title/Abstract])) OR (Treatment, Acupuncture[Title/Abstract])) OR (Therapy, Acupuncture[Title/Abstract])) OR (Pharmacoacupuncture Treatment[Title/Abstract])) OR (Treatment, Pharmacoacupuncture[Title/Abstract])) OR (Pharmacoacupuncture Therapy[Title/Abstract])) OR (Therapy, Pharmacoacupuncture[Title/Abstract])) OR (Acupotomy[Title/Abstract])) OR (Acupotomies[Title/Abstract])) OR (Acupuncture Points[Title/Abstract])) OR (Acupuncture Point[Title/Abstract])) OR (Point, Acupuncture[Title/Abstract])) OR (Points, Acupuncture[Title/Abstract])) OR (Acupoints[Title/Abstract])) OR (Acupoint[Title/Abstract]) |
| #6 | #4OR#5 |
| #7 | (((((randomized controlled trial[Publication Type]) OR (controlled clinical trial[Publication Type])) OR (randomized[Title/Abstract])) OR (placebo[Title/Abstract])) OR (randomly[Title/Abstract])) OR (trial[Title/Abstract])) OR (groups[Title/Abstract]) |
| #8 | #3 AND #6 AND #7 |

be included. The reasons for excluding studies will be recorded during the review process. The selection process will be according to the PRISMA flow chart shown in Fig 1.

## Data extraction

Two reviewers will separately extract data from the included studies into an Excel sheet. The extracted information will consist of the first author, publication year, study design, sample size, characteristics of participants, diagnostic criteria, type of interventions, treatment courses, follow-up, outcome measures, and adverse events. If there are differences in the cross-checking process, we will invite the third reviewer to mediate and reach a consensus.

## Risk of bias assessment

Two reviewers will use the Cochrane tool to independently assess the risk of bias in the included RCT (RoB 2) [34]. Based on the overall bias, studies will be assessed and classified into low, high risk of bias, or some concerns. Any differences between the investigators will be discussed. If necessary, we will consult the third investigator.

## Measures of treatment effect

We will use the relative risk with the respective 95% confidence intervals(95% CIs) to estimate categorical data. We will also use weighted or standardized mean differences to estimate continuous data with 95% CI.

## Data synthesis

The Review Manager (version 5.4) software of the Cochrane Collaboration will be used for data analysis. We will use $I^2$ statistics to assess the statistic heterogeneity in the meta-analysis. If the $I^2$ value is lower than 50% among studies, it will be considered homogeneity and utilize a fixed-effects model. If the $I^2$ value is greater than 50%, we will use a random-effect model.

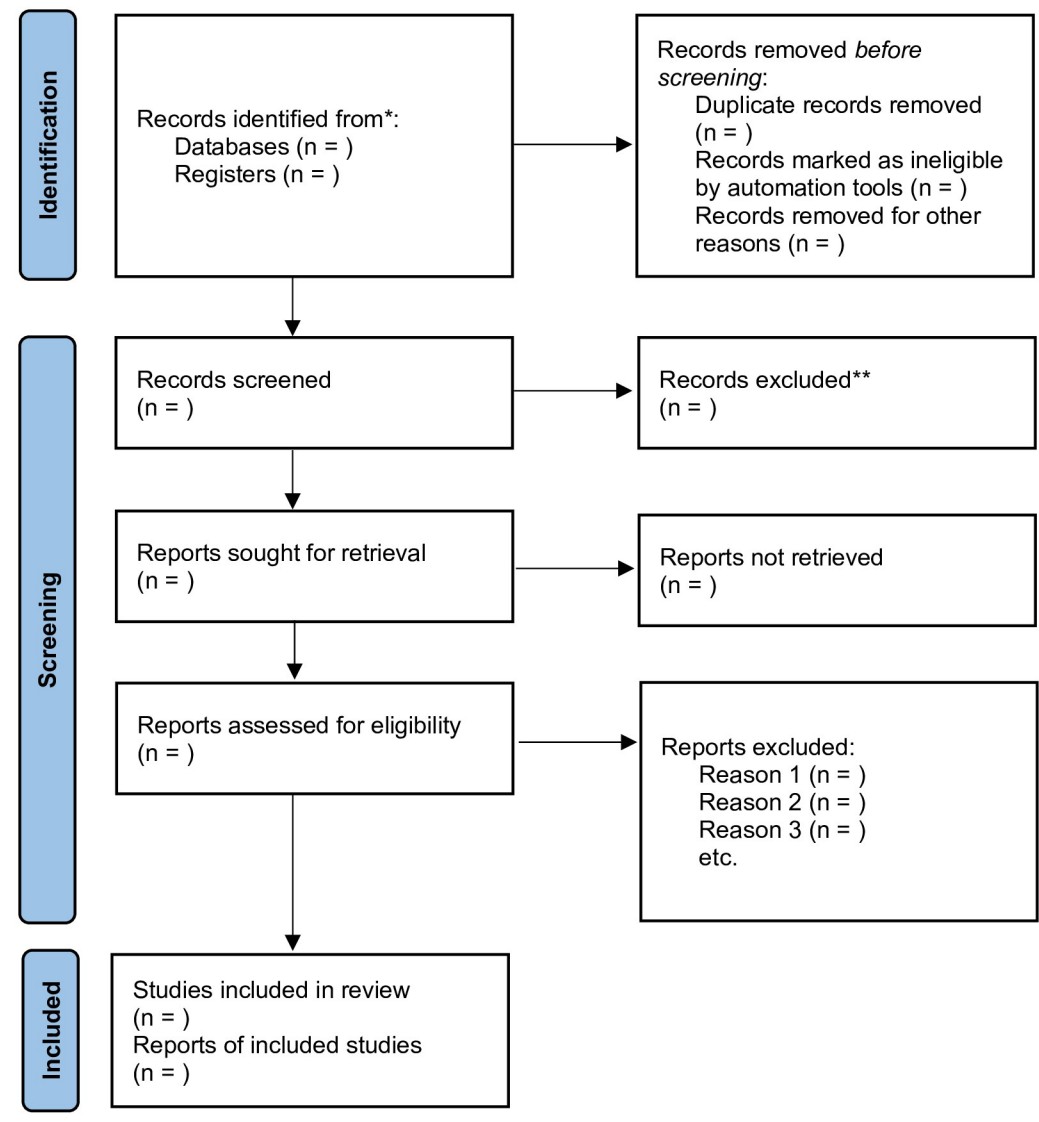

**Fig 1. Flow diagram of study selection.**

A stratified meta-analysis was conducted to evaluate the effect of different treatment courses of Acupuncturists: 2weeks, 4weeks(or 1 months), 6 weeks, and 8 weeks. If there is clinical and methodological heterogeneity, we will conducted subgroup and sensitivity analysis explore the reasons of the significant heterogeneity. We will present a narrative synthesis when heterogeneity is too apparent to resolve.

**Dealing with missing data.**   We will contact the original author to obtain relevant information when the required data is missing. If research data is unavailable, we will exclude the studies from the data synthesis.

**Subgroup analysis.**   If the clinical and methodological heterogeneity is substantial among the studies, we will attempt to conduct the subgroup analyses, which include age, gender, outcome styles, etc.

**Sensitivity analysis.**   The robustness of the study will be assessed by sensitivity analysis. After removing literature with relatively poor quality, the meta-analysis will be performed

again to compare whether there is a significant difference between the combined effect before and after.

## Meta-biases

If this review includes more than 10 studies, we will assess reporting bias using funnel plots. We will perform Egger's test to evaluate publication bias.

## Grading the quality of evidence

According to the Grading of Recommendations Assessment, Development and Evaluation (GRADE), the quality of evidence will be independently evaluated and classified as high, moderate, low, or very low by two reviewers [35].

## Ethics and dissemination

An ethical statement is not required for this review. The findings will be disseminated through peer-reviewed publications.

## Patient and public involvement

Patients and/or the public were not involved in the design, or conduct, or reporting or dissemination plans of this research.

## Discussion

The pathophysiology of IBS remains incompletely understood. It is difficult to find an effective targeted treatment. Therefore, the current treatments is often far from satisfaction in all patients [36].

Most IBS [17] patients with poor efficacy of conventional treatment seek acupuncture as an alternative treatment for poor efficacy of conventional treatment. Acupuncture can improve IBS-D by regulating multiple mechanisms such as brain-gut peptides, inflammation and hypersensitivity of bowels [20]. Evidence-based medical research suggests that acupuncture may be a promising treatment for IBS-D [18, 19].

Acupuncture efficacy depends on many aspects, including acupoint selection, acupuncture courses, depth, and current intensity. The effects of acupuncture may be influenced by different time courses in patients of IBS-D.

Nonetheless, no systematic review comparing various acupuncture courses has ever been published. Through meta-analysis, this systematic review will compare the curative effects of various acupuncture treatment courses on IBS-D. We hope that this review will provide a reference for clinical decisions made by acupuncturists.

## Supporting information

**S1 Checklist. PRISMA-P 2015 checklist.**
(PDF)

## Author Contributions

**Conceptualization:** Zhigang Li.

**Data curation:** Ting Li, Jun Zhao, Da Li.

**Formal analysis:** Junjian Tian, Jun Zhao, Jingwen Sun.

**Funding acquisition:** Zhigang Li.

**Investigation:** Rongxing Shi.

**Methodology:** Junjian Tian, Jingwen Sun, Rongxing Shi.

**Project administration:** Junjian Tian.

**Supervision:** Zhigang Li, Rongxing Shi.

**Validation:** Rongxing Shi.

**Visualization:** Rongxing Shi.

**Writing – original draft:** Junjian Tian.

**Writing – review & editing:** Junjian Tian, Ting Li, Zhigang Li.

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
