## [Decision Letter · Decision Letter 0]

24 Sep 2023

PONE-D-23-13895Efficacy of Different Courses of Acupuncture for Diarrhea Irritable bowel syndrome: a protocol for systematic review and meta-analysisPLOS ONE

Dear Dr. tian,

Thank you for submitting your manuscript to PLOS ONE. After careful consideration, we feel that it has merit but does not fully meet PLOS ONE’s publication criteria as it currently stands. Therefore, we invite you to submit a revised version of the manuscript that addresses the points raised during the review process.

The manuscript has been evaluated by three reviewers, and their comments are available below. 

The reviewers have made several requests, mostly for additional details and clarification.In addition, please could you 1) Thoroughly copyedit your manuscript for language usage, spelling, and grammar.2) Provide line numbers in your PRISMA-P checklist, specifying the location of the relevant information in the manuscript,Could you please revise the manuscript to carefully address the concerns raised?==============================

We look forward to receiving your revised manuscript.

Kind regards,

Steve Zimmerman, PhD

Associate Editor, PLOS ONE

Journal Requirements:

Reviewers' comments:

Reviewer's Responses to Questions

**Comments to the Author**

1. Does the manuscript provide a valid rationale for the proposed study, with clearly identified and justified research questions?

Reviewer #1: Yes

Reviewer #2: Yes

Reviewer #3: Yes

2. Is the protocol technically sound and planned in a manner that will lead to a meaningful outcome and allow testing the stated hypotheses?

Reviewer #1: Yes

Reviewer #2: Yes

Reviewer #3: Yes

3. Is the methodology feasible and described in sufficient detail to allow the work to be replicable?

Reviewer #1: Yes

Reviewer #2: Yes

Reviewer #3: Yes

4. Have the authors described where all data underlying the findings will be made available when the study is complete?

Reviewer #1: Yes

Reviewer #2: Yes

Reviewer #3: No

5. Is the manuscript presented in an intelligible fashion and written in standard English?

Reviewer #1: Yes

Reviewer #2: Yes

Reviewer #3: Yes

6. Review Comments to the Author

You may also provide optional suggestions and comments to authors that they might find helpful in planning their study.

Reviewer #1: The protocol is well written and meets all the methodological requirements for the conduct an adequate systematic review and meta-analysis.

I have only two considerations to make:

- It is essential for the authors to add information in the data synthesis section about how the results will be presented if conducting the meta-analysis is not possible.

- I suggest the authors use the Rayyan tool for the exclusion of duplicates; and selection of studies. Free tool designed exclusively for conducting systematic reviews.

Reviewer #2: The authors present a systematic review protocol whose objective is to evaluate effectiveness of Acupuncture of different courses in the treatment of IBS-D and provide sufficient evidence for clinical recommendations for IBS-D. The manuscript was very well written and contemplates what is recommended in PRISMA-P.

Only , I suggest clarification on which validated tools should have been used in the RCTs to assess the primary and secondary outcomes defined by the authors.

"Efficiency rate and the Symptoms Severity Score", which are the primary ones, probably have validated instruments and I suggest that you include the main ones with references. Likewise, there are validated instruments for abdominal pain scores, abdominal frequency scores, satisfaction with bowel habits scores and quality of life and the authors must mention which ones will be considered and cite their references in the manuscript.

Reviewer #3: In general, the paper is well-written and logically presented. However, there are a few points that require your attention：

1.Inclusion criteria

1.1 Diagnostic criteria for IBS-D lack references.

1.2 Scoring criteria for outcome indicators were not explained.

2.Many formatting errors in this article.

3. The title is "Efficacy of Different Courses of Acupuncture for Diarrhea Irritable bowel syndrome: a protocol for systematic review and meta-analysis", but the inclusion criteria is "acupuncture or EA", please explain.

7. PLOS authors have the option to publish the peer review history of their article (what does this mean?). If published, this will include your full peer review and any attached files.

Reviewer #1: No

Reviewer #2: **Yes: **Ricardo Ney Cobucci

Reviewer #3: No

---

## [Author Response · Author response to Decision Letter 0]

23 Oct 2023

We have revised the manuscript which is marked in red or bule in the paper, according to the comments and suggestions of reviewers.

1.Dear editors. We appreciate the Editor’s comment.

 1.1We've carefully checked the language usage, spelling, and grammar of the manuscript.

 1.2We have provided the line numbers for the relevant information in my PRISMA-P checklist

2.Dear reviewer#1, thank you for your comments.

Dear reviewer, We agree with the reviewer’s comment..

 2.1Thank you very much for your advice! I will be adding relevant information to the article.We will present a narrative synthesis when heterogeneity is too apparent to resolve.

 2.2EedNote software has the function of automatically finding duplicates, and we will also carry out manual exclusion of duplicates to ensure the quality of the literature.

3.Dear reviewer#2, We acknowledge the reviewer’s comment.

 Thank you for your advice! I will add the relevant content and apply the validated tools to define validity. The details are as follows:

 The primary outcome measures of this review will be the efficiency rate and the Symptoms Severity Score [1]. The clinical efficiency rate was defined as the degree of relief of IBS symptoms, which are assessed through the Symptoms Severity Score. This includes the use of the IBS Symptom Severity Scale (IBS-SSS)[2], or the Gastrointestinal Symptom Rating Scale for IBS (GSRS-IBS)[3]. 

4.Dear reviewer#3, We appreciate the reviewer’s comment.

 4.1Inclusion criteria

 4.1.1 I will add references to the diagnostic criteria for IBS-D.The patients diagnosed with IBS-D will be eligible based on Rome IV/III/II criteria[4-7].

 4.1.2 I will add the relevant content and apply the validated tools to define validity. The details are as follows:

 The primary outcome measures of this review will be the efficiency rate and the Symptoms Severity Score [1]. The clinical efficiency rate was defined as the degree of relief of IBS symptoms, which are assessed through the Symptoms Severity Score. This includes the use of the IBS Symptom Severity Scale (IBS-SSS)[2], or the Gastrointestinal Symptom Rating Scale for IBS (GSRS-IBS)[3]. 

 4.2 I'm going to carefully revise the formatting of my post with a view to achieving good results.

 4.3 Acupuncture, including manual acupuncture (also known as Acupuncture) and electroacupuncture, is a commonly used therapeutic technique in clinical practice. Most meta-analysis studies on acupuncture have included electroacupuncture as an intervention.

5.Dear reviewer#3, thank you for your comments.

 We appreciate the reviewer’s comment. In the submission system, I have described that the relevant data will be made public after the study is completed.

We would like to resubmit this revised manuscript to “PLOS ONE ” and hope that it will be acceptable for publication in the journal.

References

1. Wang M, Xie X, Zhao S, Ma X, Wang Z, Zhang Y. Fecal microbiota transplantation for irritable bowel syndrome: a systematic review and meta-analysis of randomized controlled trials. Frontiers in immunology. 2023;14:1136343.

2. Francis CY, Morris J, Whorwell PJ. The irritable bowel severity scoring system: a simple method of monitoring irritable bowel syndrome and its progress. Alimentary pharmacology & therapeutics. 1997;11(2):395-402.

3. Wiklund IK, Fullerton S, Hawkey CJ, Jones RH, Longstreth GF, Mayer EA, et al. An irritable bowel syndrome-specific symptom questionnaire: development and validation. Scandinavian journal of gastroenterology. 2003;38(9):947-54.

4. Badia X, Mearin F, Balboa A, Baro E, Caldwell E, Cucala M, et al. Burden of illness in irritable bowel syndrome comparing Rome I and Rome II criteria. PharmacoEconomics. 2002;20(11):749-58.

5. Markert C, Suarez-Hitz K, Ehlert U, Nater UM. Endocrine dysregulation in women with irritable bowel syndrome according to Rome II criteria. J Behav Med. 2016;39(3):519-26.

6. Drossman DA, Dumitrascu DL. Rome III: New standard for functional gastrointestinal disorders. Journal of gastrointestinal and liver diseases : JGLD. 2006;15(3):237-41.

7. Barberio B, Houghton LA, Yiannakou Y, Savarino EV, Black CJ, Ford AC. Symptom Stability in Rome IV vs Rome III Irritable Bowel Syndrome. The American journal of gastroenterology. 2021;116(2):362-71.

---

## [Decision Letter · Decision Letter 1]

15 Nov 2023

Efficacy of Different Courses of Acupuncture for Diarrhea Irritable bowel syndrome: a protocol for systematic review and meta-analysis

PONE-D-23-13895R1

Dear Dr. tian,

We’re pleased to inform you that your manuscript has been judged scientifically suitable for publication and will be formally accepted for publication once it meets all outstanding technical requirements.

Kind regards,

Yuzhen Xu

Academic Editor

PLOS ONE

Additional Editor Comments (optional):

Congratulations, your manuscript has met the criteria for publication in PLOS ONE.

Reviewers' comments:

Reviewer's Responses to Questions

**Comments to the Author**

1. Does the manuscript provide a valid rationale for the proposed study, with clearly identified and justified research questions?

Reviewer #1: Yes

Reviewer #2: Yes

Reviewer #3: Yes

2. Is the protocol technically sound and planned in a manner that will lead to a meaningful outcome and allow testing the stated hypotheses?

Reviewer #1: Yes

Reviewer #2: Yes

Reviewer #3: Yes

3. Is the methodology feasible and described in sufficient detail to allow the work to be replicable?

Reviewer #1: Yes

Reviewer #2: Yes

Reviewer #3: Yes

4. Have the authors described where all data underlying the findings will be made available when the study is complete?

Reviewer #1: Yes

Reviewer #2: Yes

Reviewer #3: Yes

5. Is the manuscript presented in an intelligible fashion and written in standard English?

Reviewer #1: Yes

Reviewer #2: Yes

Reviewer #3: Yes

6. Review Comments to the Author

You may also provide optional suggestions and comments to authors that they might find helpful in planning their study.

Reviewer #1: The article addresses an interesting topic and is well written, following all the methodological rules that characterize the type of study. In addition, the authors followed all comments and suggestions of reviewers.

Reviewer #2: The authors met the reviewers' recommendations and the revised manuscript can be published. Congratulations.

Reviewer #3: The article has a complete structure and innovative content, which can be considered for publication.

7. PLOS authors have the option to publish the peer review history of their article (what does this mean?). If published, this will include your full peer review and any attached files.

Reviewer #1: No

Reviewer #2: **Yes: **Ricardo Ney Cobucci

Reviewer #3: No

---

## [Editor Report · Acceptance letter]

5 Dec 2023

PONE-D-23-13895R1 

Efficacy of Different Courses of Acupuncture for Diarrhea Irritable bowel syndrome: a protocol for systematic review and meta-analysis 

Dear Dr. tian:

I'm pleased to inform you that your manuscript has been deemed suitable for publication in PLOS ONE. Congratulations! Your manuscript is now with our production department. 

Kind regards, 

on behalf of

Professor Yuzhen Xu 

Academic Editor

PLOS ONE